# Angular and Shell-Aware Deep Potential Energy Model for Molecular Dynamics

## Abstract

Angular information, especially involving the first and second coordination shells, is critical for accurately describing the potential energy surface (PES) in molecular systems. However, existing machine learning PES models either neglect this information or indiscriminately process it from all neighbors, blurring the critical contributions of distinct shells and compromising their predictive accuracy. In this work, we propose the Angular and Shell-Aware Deep Potential (ASDP), a novel architecture designed to overcome this limitation. Based on the DPA-1 attention mechanism, ASDP integrates a specialized encoding module that selectively processes angular information confined within the first two coordination shells. This shell-aware approach allows for a more physically meaningful representation of the local atomic environment. Experimental results show that by capturing crucial shell-specific angular dependencies, ASDP represents the PES of various molecular systems with the *ab initio* quantum mechanics (QM) accuracy, outperforming many existing methods and offering a new direction for creating highly accurate and robust machine learning potentials. Our code can be found in `https://anonymous.4open.science/r/ASDP-ICLR-code`.

## 1 Introduction

Molecular dynamics (MD) simulations are an indispensable tool in modern science, offering atomic-scale insights into processes across chemistry, materials science, and biology Allen & Tildesley (2017); Karplus & McCammon (2002). However, the field has long been defined by a fundamental trade-off between accuracy and efficiency. Ab initio molecular dynamics (AIMD), based on quantum mechanics, provides benchmark accuracy but its prohibitive computational cost restricts simulations to small systems and short timescales Car & Parrinello (1985); Marx & Hutter (2000). Conversely, classical force fields enable large-scale simulations, but their reliance on predefined analytical functions limits their accuracy and transferability, particularly for describing complex chemical environments or reactive events Ponder & Case (2003). This persistent gap between speed and fidelity has hindered the simulation of many large-scale, complex phenomena.

Machine learning potentials (MLPs) promise *ab initio* accuracy at a fraction of the computational cost by learning the potential energy surface (PES) from quantum calculations Behler & Parrinello (2007); Wang et al. (2018). Their predictive power hinges on an accurate representation of the local atomic environment, for which many-body angular information is a prerequisite. Crucially, this geometrically-decisive information is concentrated within the first two coordination shells. The first shell's angularity defines the strong, directional interactions that dictate local geometry, while the second shell governs the more subtle forces (e.g., torsional and van der Waals) that are nonetheless critical for determining bulk properties like phase stability and conformational dynamics. Therefore, a high-fidelity MLP must effectively capture the angular contributions from both shells to construct an accurate and comprehensive PES.

The necessity of accurately capturing angular information has driven the development of diverse MLP architectures. Among these, descriptor-based models like BPNN Behler & Parrinello (2007), ANI Smith et al. (2017) and AIMNet Zubatyuk et al. (2019) explicitly encode the angular relationships. The current state-of-the-art in this class, DPA-1 Zhang et al. (2024), leverages a powerful attention mechanism for remarkable efficiency and transferability. However, its handling of angular information relies on a heuristic fusion within the attention mechanism; by imposing a

rigid and simplistic prior on how angular and radial data interact, this approach can lead to an imprecise representation of complex geometric dependencies. Other major architectural families tackle angular information differently: some message-passing neural networks (MPNNs) capture it implicitly Schütt et al. (2018); Unke & Meuwly (2019); Qiao et al. (2020); Schütt et al. (2021), meanwhile variants such as DimeNet/DimeNet++ Gasteiger et al. (2020b;a) add explicit angular terms, and recent equivariant networks like NequIP Batzner et al. (2022), Allegro Musaelian et al. (2023) and MACE Batatia et al. (2022) employ spherical harmonics for a complete basis. Yet, despite these varied and sophisticated strategies, a more fundamental physical distinction is commonly overlooked: **These models treat all angular information monolithically, failing to prioritize the geometrically-crucial contributions from the first two coordination shells over the noise from distant, uncorrelated atoms**. This failure to filter out angular noise, which is less essential for an accurate PES construction, inherently limits the model's physical realism and predictive fidelity.

To overcome these challenges, we introduce the Angular and Shell-aware Deep Potential (ASDP). Its key innovation is a dedicated angular encoding module that injects geometric information directly into the attention mechanism as an additive bias to the pre-softmax logits. This makes local geometry an intrinsic component of the attention calculation, rather than an external, post-hoc multiplicative filter as employed in models like DPA-1. Crucially, ASDP enforces a strong physical prior by constraining this angular encoding to the first two coordination shells. This design isolates the critical angular 'signal'—spanning both directional bonds and proximal non-bonded effects—from the 'noise' of distant, uncorrelated neighbors. By embedding a physically-grounded angular bias within this shell-aware framework, ASDP constructs a more robust and physically-plausible atomic representation, achieving superior accuracy and robustness.

The major contributions of this paper are summarized as follows:

- We identify two critical flaws in state-of-the-art attention-based deep PES models: their multiplicative angular mechanism can erroneously nullify key chemical interactions, and they indiscriminately include noisy, long-range angular information.

- We propose the Angular and Shell-Aware Deep Potential (ASDP), a novel architecture that resolves these flaws by integrating a learnable, additive angular bias within a physically-motivated framework constrained to the first two coordination shells.

- Our resulting ASDP model achieves state-of-the-art or highly competitive accuracy in energy and force prediction across a diverse suite of six molecular systems, validating the superior performance and transferability of our design.

## 2 Preliminaries

### 2.1 Descriptor-Based Deep PES Model

Descriptor-based models represent a foundational paradigm for constructing machine learning potentials, built upon the physically-motivated locality assumption Behler & Parrinello (2007). This principle decomposes the total potential energy of a system, $E_{\text{total}}$, into a sum of individual atomic energy contributions, $E_i$, where each atomic energy $E_i$ is determined solely by its local chemical environment, $R_i$, defined as the set of neighboring atoms within a cutoff radius $r_{\text{cut}}$. This decomposition makes the model scalable to large systems.

To enforce fundamental physical symmetries, this approach utilizes a two-component architecture that guarantees total energy ($E_{\text{total}}$) invariance and atomic force ($\mathbf{F}_i$) equivariance. The process involves two key stages: 1) a **descriptor** ($D_i$) transforms the raw local environment coordinates ($R_i$) into a fixed-size feature vector that is explicitly designed to be invariant to translation, rotation, and the permutation of identical atoms; and 2) a **fitting network** maps this invariant descriptor to a scalar atomic energy $E_i$. Because the input to the network is inherently invariant, the output energy is also guaranteed to be invariant.

The atomic forces $\mathbf{F}_i$, crucial for molecular dynamics, are obtained analytically as the negative gradient of the total energy with respect to atomic positions $\mathbf{r}_i$. As both the descriptor and the

network are differentiable, forces can be computed efficiently via the chain rule:

$$\mathbf{F}_i = -\nabla_{\mathbf{r}_i} E_{\text{total}} = -\sum_j \frac{\partial E_{\text{total}}}{\partial D_j} \frac{\partial D_j}{\partial \mathbf{r}_i}. \tag{1}$$

This analytical differentiation mathematically guarantees that the forces are correctly equivariant, allowing the model to adhere to physical symmetries by construction.

## 2.2 Physical Significance of Angular Information and Coordination Shells

The potential energy of an atom is dictated by its local geometry, which can be decomposed into the radial distribution of neighbors and their relative angular arrangement. However, the physical significance of this angular information is not uniform but is instead highly distance-dependent. This importance is most pronounced within the first coordination shell, where the formation of strong, directional chemical bonds defines the core molecular geometry, thereby exerting the most direct and powerful influence on the system's energy and atomic forces.

Moving outward, the second coordination shell is governed by weaker, non-bonded interactions, yet its angular structure remains crucial for resolving subtle but decisive many-body effects. For instance, it is essential for distinguishing between energetically similar crystal structures like FCC and HCP Kittel & McEuen (2018), and for defining the torsional angles that control conformational energy landscapes in molecular systems. Beyond this critical sphere, the rapid decay of angular influence physically justifies a finite cutoff radius, as contributions from distant, angularly-uncorrelated atoms become negligible Behler & Parrinello (2007).

## 3 Methodology

### 3.1 DPA-1: Mechanism & Limitations

**Denotation.** Given a system of $N$ atoms with element types $\mathcal{A} = \{a_i, a_2, ..., a_N\}$ and Cartesian coordinates $\mathcal{R} = \{r_1, r_2, ..., r_N\}$. For each atom $i$, denote its neighbors by $\{j | j \in \mathcal{N}_{r_c}(i)\}$, where $\mathcal{N}_{r_c}(i)$ denotes the set of atom indices such that $|r_{ji}| < r_c$ and $|r_{ji}|$ is the Euclidean distance between atoms $i$ and $j$, and $r_c$ is the cutoff radius. Let $N_i = |\mathcal{N}_{r_c}(i)|$ and denote $R^i \in \mathbb{R}^{N_i \times 3}$ as the neighboring coordinates relative to $i$ (row $j$ in $R^i$, which is $\{x_{ji} = x_j - x_i, y_{ji} = y_j - y_i, z_{ji} = z_j - z_i\}$, can be denoted as $r_{ji}$), the task of DPA-1 is to map element types in $\mathcal{N}_{r_c}(i)$ and $R^i$ to atomic energy $E_i$ through the self-attention-based descriptor and the fitting network.

**Mechanism.** Figure 1.(a) shows the overall flow of DPA-1: For an atom $i$ with neighbor set $\mathcal{N}_{r_c}(i)$ and the relative coordinates $R_i$, DPA-1 starts with transforming $R^i$ into the generalized coordinates $\tilde{R}^i \in \mathbb{R}^{N_i \times 4}$, where each row $\{x_{ji}, y_{ji}, z_{ji}\}$ in $R^i$ is mapped to a row $\{\hat{x}_{ji} = \frac{s(r_{ji})x_{ji}}{|r_{ji}|}\}, \hat{y}_{ji} = \frac{s(r_{ji})y_{ji}}{|r_{ji}|}\}, \hat{z}_{ji} = \frac{s(r_{ji})z_{ji}}{|r_{ji}|}, s(r_{ji})$ in $\tilde{R}^i$, where $s(r_{ji}) : \mathbb{R} \mapsto \mathbb{R}$ is a continuous and differentiable smoothing function. Next, each atom element type $a_i$ is mapped by the type embedding net into the one-hot-like high-dimensional representation $T_i$. Subsequently, using the radial information $\tilde{R}^T$ and type embeddings $\{T_i\} \cup \{T_j | j \in \mathcal{N}_{r_c}(i)\}$, we obtain the local embedding matrix $\mathcal{G}^i \in \mathbb{R}^{N_i \times M_1}$ via a neural network $G$. Each row $(\mathcal{G}^i)_j$ is computed as $G(s(r_{ji})||T_i||T_j)$, where $||$ denotes concatenation and $M_1$ is the embedding dimension.

The core of DPA-1 is a multi-layer self-attention modules. As is shown in Figure 1.(b), this module refines the initial embedding $\mathcal{G}^i$ by re-weighting interactions among neighbors. In each attention layer $l$, starting with the initial embedding $\mathcal{G}^i$ (note that $\mathcal{G}^{i,0} = \mathcal{G}^i$), the input representation $\mathcal{G}^{i,l-1}$ is first linearly projected to produce queries ($Q^{i,l}$), keys ($K^{i,l}$), and values ($V^{i,l}$), all of dimension $\mathbb{R}^{N_i \times d}$. Subsequently, an attention weight matrix $\varphi$ is calculated, which uniquely combines both distance-based (via dot-product) and angular information:

$$\varphi(Q^{i,l}, K^{i,l}, R^{i,l}) = \text{softmax}\left(\frac{Q^{i,l}(K^{i,l})^T}{\sqrt{d}}\right) \odot \hat{R}^i(\hat{R}^i)^T. \tag{2}$$

In this formulation, $\hat{R} = \frac{R^i}{||R^i||_2} \in \mathbb{R}^{N_i \times 3}$, and the standard scaled dot-product attention is enhanced by an element-wise multiplication ($\odot$) with the matrix $\hat{R}^i(\hat{R}^i)^T$. **This matrix, containing the**

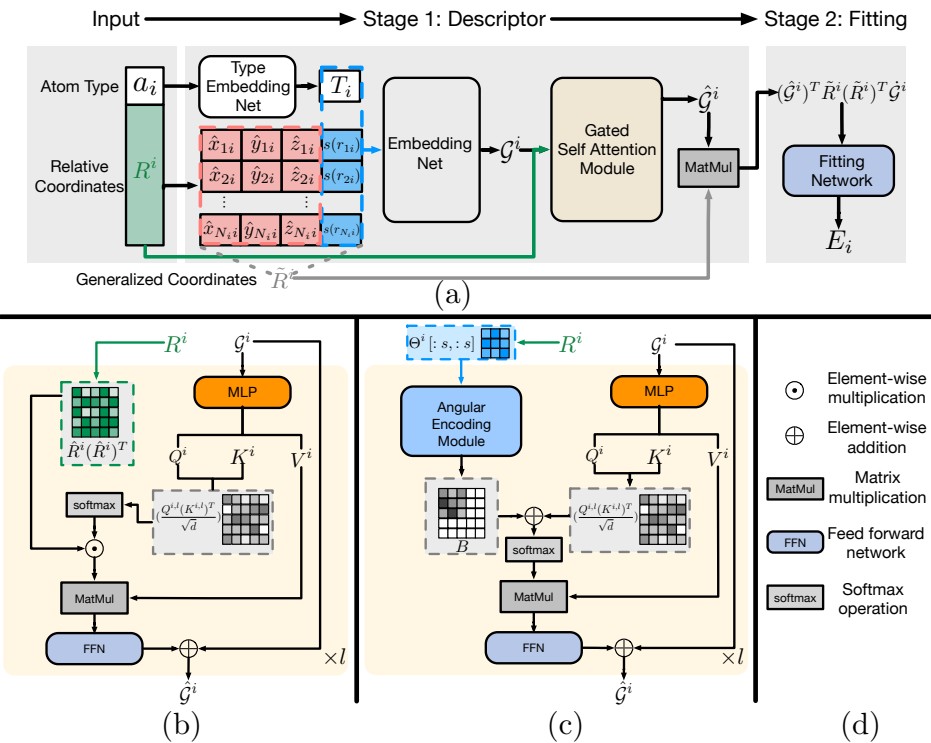

Figure 1: (a) Overall model architecture of DPA-1 and ASDP, (b) self-attention module of DPA-1, (c) angular and shell-aware self attention module for ASDP and (d) the corresponding legends.

**pairwise cosine similarities of the neighbor direction vectors, directly injects geometric angular information into the attention mechanism**. The layer's output is then computed using a residual connection and Layer Normalization, producing the updated representation $\mathcal{G}^{i,l}$:

$$\mathcal{G}^{i,l} = \mathcal{G}^{i,l-1} + \text{LayerNorm}(A(Q^{i,l}, \mathcal{K}^{i,l}, \mathcal{V}^{i,l}, \mathcal{R}^{i,l})). \tag{3}$$

After the final attention layer, the refined representation $\hat{\mathcal{G}}^i$ is used to construct a rotationally invariant descriptor matrix $\mathcal{D}^i \in \mathbb{R}^{M_1 \times M_2}$. This descriptor is defined as:

$$\mathcal{D}^i = (\hat{\mathcal{G}}^i)^T \tilde{\mathcal{R}}^i (\tilde{\mathcal{R}}^i)^T \dot{\mathcal{G}}^i, \tag{4}$$

where $\dot{\mathcal{G}}^i$ is a designated sub-matrix of $\hat{\mathcal{G}}^i$. Finally, the descriptor $\mathcal{D}^i$ is concatenated with the central atom's type embedding $T_i$ and fed into a fitting network to predict the atomic energy $E_i$.

**Limitations.** While DPA-1 incorporates angular information through equation 2, its direct implementation introduces two significant limitations. First, the re-weighting of the pairwise attention matrix $\text{softmax}(\frac{Q^{i,l}(K^{i,l})^T}{\sqrt{d}})$ via an element-wise multiplication with a matrix of cosine values (derived from neighbor vectors $\tilde{R}^i(\tilde{R}^i)^T$, as shown in Equation.2) creates a critical flaw: **the re-weighting factor is directly and linearly dependent on the cosine value, a rigid formulation that lacks the flexibility to capture the true, non-linear chemical importance of specific angular configurations**. This is illustrated by the phosphorus pentachloride ($PCl_5$) system shown in Figure 2. In its trigonal bipyramidal (TBP) structure, both the 90° axial-equatorial (Ax-Eq) and 120° equatorial-equatorial (Eq-Eq) angles are chemically decisive. The Ax-Eq repulsion makes the axial bonds longer and weaker, while the Eq-Eq interactions define the stability of the equatorial plane. However, the DPA-1 mechanism nullifies the contribution of these 90° interactions by assigning them a re-weighting factor of

Figure 2: Bond angles of the $PCl_5$ system.

zero $(\cos(90°)=0)$, while assigning a non-zero weight to the $120°$ interactions $(\cos(120°)=-0.5)$. Consequently, the model becomes "blind" to the primary sources of steric strain, leading to a flawed PES that fails to capture the high energy cost of the molecule's steric repulsions.

A second limitation stems from the indiscriminate inclusion of pairwise angular information from all neighbors of a central atom $i$. As mentioned in Section. 1 and 2.2, angular terms critical for defining molecular structure and energy are typically confined to the first and second coordination shells. By incorporating angular data from beyond these local shells, DPA-1 introduces a significant volume of redundant and potentially noisy information. This increases the computational burden without a corresponding improvement in performance and may even degrade the model's accuracy by obscuring the chemically relevant local interactions.

### 3.2 ASDP: OVERVIEW

To address the limitations of DPA-1 in its direct incorporation of angular information for PES modeling, we propose the Angular and Shell-aware Deep Potential (ASDP) model. Our model is built upon the descriptor-based architecture adopted by DPA-1 (shown in Figure 1.(a)), ensuring a robust foundation for representing local atomic environments. The primary contribution of ASDP lies in a redesigned self-attention module that achieves two key objectives: first, it introduces a more sophisticated mechanism for integrating angular information, and second, it explicitly accounts for the influence of distinct coordination shells on these geometric relationships.

The operational workflow of the ASDP attention module is illustrated in Figure 1.(c). The process begins, analogous to DPA-1, by generating a local representation matrix, $\mathcal{G}^i$, for a central atom $i$, which is then projected through MLPs to produce queries $(Q^{i,l})$, keys $(K^{i,l})$, and values $(V^{i,l})$. From these, we compute the initial distance-based attention scores, $\frac{Q^{i,l}(K^{i,l})^T}{\sqrt{d}}$. Concurrently, a separate branch processes angular information in a shell-aware manner. **For atom $i$, we denote the total number of neighbors within its first and second coordination shells as $s$.** After obtaining the angle matrix $\Theta^i[:s,:s]$ which stores the angle of each triplet atoms (the angle is subtended at the central atom $i$) among the $s$ neighbors, each element $\theta^i_{j,k}$ in the matrix is passed through an angular encoding module to yield a scalar value, $b_{j,k}$. These scalars form a bias matrix $B^{\text{shell}} \in \mathbb{R}^{s \times s}$, which exclusively captures the angular interactions within the first two coordination shells. To align this shell-specific bias with the full $N_i \times N_i$ attention score matrix (where $N_i$ is the total number of neighbors), we construct the final bias matrix, $B$, by embedding $B^{\text{shell}}$ into an $N_i \times N_i$ zero matrix. This operation ensures that only neighbors within the specified shells contribute an angular bias, and can be formally expressed as:

$$B_{j,k} = b_{j,k} \cdot \mathbb{I}(j \leq s \wedge k \leq s), \tag{5}$$

where $\mathbb{I}(\cdot)$ is the indicator function. This full bias matrix $B$ is then introduced as an attention bias, directly added to the initial distance-based scores to obtain the definitive angular and shell-aware attention weights:

$$\varphi(Q^{i,l}, K^{i,l}, R^{i,l}) = \text{softmax}\left(\frac{Q^{i,l}(K^{i,l})^T}{\sqrt{d}} + B\right). \tag{6}$$

The remaining process of obtaining $\hat{\mathcal{G}}^i$ and $\mathcal{D}^i$ is similar to Equation 3 and 4.

Our approach presents two key advantages over DPA-1. First, ASDP integrates angular information as a learned additive bias before softmax. This avoids the DPA-1 method of directly multiplying attention weights with raw cosine values, which can misleadingly suppress interactions, and instead provides a more robust and flexible modulation of atomic relationships. Second, the model's explicit shell aware focus applies this angular correction only to the most chemically relevant neighbors, introducing a physically meaningful inductive bias that the uniform treatment in DPA-1 lacks.

### 3.3 ASDP: SHELL-AWARE ANGULAR ENCODING MODULE

The Shell-Aware Angular Encoding Module is architected as a specialized sub-module that dynamically generates a scalar attention bias by interpreting the three-body geometry of the local atomic environment, and its architecture and workflow are illustrated in Figure 3. Given the central atom $i$ along with its $N_i$ neighbors, we first identify the neighbor subset residing within the first and second

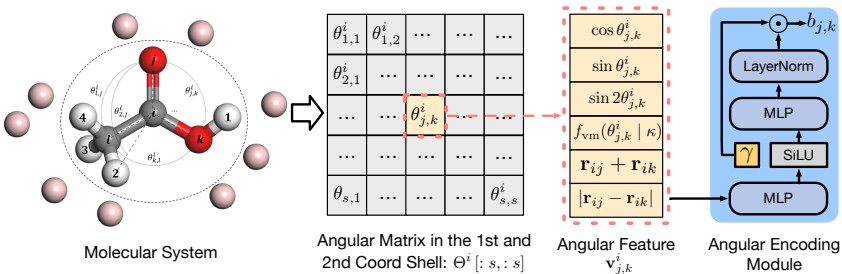

Figure 3: Architecture and workflow of the Shell-Aware Angular Encoding Module.

coordination shells, denoting the count of these neighbors as $s$. The module's focus is then exclusively directed towards the angular information derived from triplets $(j, i, k)$, where both $j$ and $k$ are members of this shell-aware set. For each such triplet, the angle $\theta_{j,k}^i$ subtended at the central atom $i$ is calculated using the relative position vectors $r_{ji}$ and $r_{ki}$:

$$\theta_{j,k}^i = \arccos\left(\frac{r_{ji} \cdot r_{ki}}{|r_{ji}||r_{ki}|}\right). \tag{7}$$

Following this, a dedicated encoding neural network maps each angle $\theta_{j,k}^i$, enriched with corresponding distance information, into a single scalar bias value $b_{j,k}$. This transformation is accomplished through a sophisticated feature engineering stage, which prepares a comprehensive input vector for a neural network mapper. Specifically, this feature vector is composed of 6 components: 1) $\cos(\theta_{j,k}^i)$ and 2) $\sin(\theta_{j,k}^i)$, which together form the most direct, orthogonal representation of the angle. This is augmented by 3) $\sin(2 \cdot \theta_{j,k}^i)$, which provides higher-frequency information and allows the model to capture more complex, explicit interaction features. Furthermore, to capture the importance of localized bond angle regions, we incorporate 4) a single feature derived from the von Mises distribution. We use the unnormalized probability density centered at zero ($\mu = 0$), whose functional form is given by:

$$f_{\mathrm{vm}}(\theta_{j,k}^i \mid \kappa) = \exp(\kappa \cos(\theta_{j,k}^i)), \tag{8}$$

where $\kappa$ is a hyperparameter that controls the concentration of the distribution and set to 2 in our experiments. Finally, to provide the module with crucial bond length information while maintaining permutation symmetry with respect to atoms $j$ and $k$, we include two distance-based features: 5) the sum of the bond lengths, $\mathbf{r}_{ij} + \mathbf{r}_{ik}$, and 6) the absolute difference between them, $|\mathbf{r}_{ij} - \mathbf{r}_{ik}|$. These six engineered features are then stacked together to form the final input vector $\mathbf{v}_{j,k}^i$ for the encoding module:

$$\mathbf{v}_{j,k}^i = \mathrm{stack}\left[\cos(\theta_{j,k}^i), \sin(\theta_{j,k}^i), \sin(2 \cdot \theta_{j,k}^i), f_{\mathrm{vm}}(\theta_{j,k}^i \mid \kappa), \mathbf{r}_{ij} + \mathbf{r}_{ik}, |\mathbf{r}_{ij} - \mathbf{r}_{ik}|\right]. \tag{9}$$

The Angular Encoding Module then processes this feature vector $\mathbf{v}_{j,k}^i$. It is implemented as a two-layer Multi-Layer Perceptron (MLP) with a SiLU activation function. The MLP is designed to map the input feature vector to a single scalar value. Let $W_1$ and $b_1$ be the weight and bias of the first layer, and $W_2$ and $b_2$ be those of the second layer. The computation proceeds as follows:

$$h = \mathrm{SiLU}(W_1 \cdot \mathbf{v}_{j,k}^i + b_1), \quad o_{\mathrm{raw}} = W_2 \cdot h + b_2. \tag{10}$$

The raw scalar output, $o_{\mathrm{raw}}$, is subsequently normalized and scaled to produce the final bias $b_{j,k}$. This process serves a dual purpose: the Layer Normalization step stabilizes the training dynamics, while the multiplication by a learnable scale factor $\gamma$ grants the model the flexibility to dynamically control the magnitude of the angular bias's influence on the final attention scores. The operation is defined as:

$$b_{j,k} = \gamma \cdot \mathrm{LayerNorm}(o_{\mathrm{raw}}). \tag{11}$$

## 4 EXPERIMENTS

### 4.1 MOLECULAR SYSTEM DATASETS

To comprehensively validate the accuracy and transferability of ASDP, we curated a benchmark suite of six diverse molecular systems, each partitioned into training and testing sets for rigorous

evaluation. The suite begins with a ternary **AlMgCu alloy** Jiang et al. (2021) to specifically test compositional transferability, using its unary and binary configurations for training and reserving the more complex ternary alloys for testing. We then address conformational complexity with the **ANI-1** dataset Smith et al. (2017), focusing on large organic molecules with over 40 atoms. The benchmark also includes a Li-based solid-state electrolyte system Huang et al. (2021) (**SSE-PBE** for short) to challenge the model's ability to capture ion diffusion dynamics amidst high substitutional disorder. The complexity is further increased with a **high-entropy carbonitride (HECN)** material Baidyshev et al. (2024); Nikitin et al. (2025), testing robustness during a solid-liquid phase transition under extreme disorder. A **2D In$_2$Se$_3$** ferroelectric Wu et al. (2021) is included to test the model's precision, as it demands resolution of minute energy barriers while describing both strong covalent and weak van der Waals forces. Finally, an **organic-reaction** dataset obtained from Transition-1x Schreiner et al. (2022) assesses the model's capacity as a reactive potential by requiring an accurate description of bond breaking and formation at transition states. Among them, AlMgCu, SSE-PBE, and 2D-In$_2$Se$_3$ were generated using DPGEN Zhang et al. (2019; 2020). Collectively, these systems establish a rigorous benchmark spanning metallic, covalent, and ionic bonding, as well as phase transitions and chemical reactions, thereby providing a thorough evaluation of ASDP's capabilities. Further details for each system are provided in the Appendix A.2.

### 4.2 MODELS

Our ASDP model is developed upon the DPA-1 code skeleton and incorporates a shell-aware angular encoding mechanism into its self-attention module. This mechanism is controlled by a key hyperparameter, $s$, which defines the number of nearest neighbors included in the angular encoding. For operational simplicity of ASDP, during all experiments, $s$ was set to a fixed value for all atoms within a given system, chosen based on a *priori* chemical and structural knowledge. Specifically, for crystalline solids with well-defined neighbor shells (AlMgCu alloy, HECN rock-salt) and the SSE-PBE electrolyte, we set $s = 18$ to encompass the first two coordination shells. In contrast, for systems dominated by covalent bonding (ANI-1, organic-reactions) or with diverse coordination environments (2D-In$_2$Se$_3$), a more conservative value of $s = 12$ was empirically selected. Further details on the selection of $s$ for each system are provided in Appendix A.2.

For a comprehensive comparative analysis, we benchmark our ASDP model against four SOTA deep learning potential models. These baselines represent two distinct architectural paradigms. The first category includes descriptor-based models featuring learnable descriptors, namely DeepPot-SE Zhang et al. (2018b) and DPA-1 Zhang et al. (2024). The second category comprises leading equivariant message-passing neural network models, including Nequip Batzner et al. (2022) and Allegro Musaelian et al. (2023). A detailed description of the hyperparameter settings and training details for both our ASDP model and all selected baseline models is provided in the Appendix A.3.

### 4.3 MAIN RESULTS

The comprehensive performance evaluation of ASDP against state-of-the-art baseline models across six diverse molecular systems is presented in Table 1. The results demonstrate that ASDP achieves state-of-the-art or highly competitive accuracy and showcases superior robustness, particularly in complex material systems where other advanced models fail.

A primary finding is the superior operational robustness of ASDP, which stands in stark contrast to leading equivariant models. As shown in Table 1, Nequip and Allegro failed to converge (N/A) on several complex systems. We define N/A as cases where, after extensive training (at least 10 epochs), the model's loss did not stabilize, resulting in a catastrophic test energy RMSE exceeding 1000 meV. This instability likely stems from the numerical challenges of high-order tensor products and the highly constrained optimization landscape inherent to their strictly equivariant architectures. In contrast, ASDP's consistent and stable convergence across all benchmarks affirms the practical reliability of its descriptor-based framework for challenging real-world scientific applications. Moreover, while Nequip yielded the most accurate force predictions for AlMgCu, SSE-PBE, and 2D-In$_2$Se$_3$, its energy RMSEs were catastrophically high, all in the three-digit meV range.

By incorporating a more physically-grounded description of interatomic interactions, ASDP achieves state-of-the-art accuracy. On the SSE-PBE solid-state electrolyte, for instance, ASDP reduces the energy RMSE by nearly 40% compared to DPA-1 (from 3.8 to 2.3 meV), a critical

| Systems | Nequip | | Allegro | | DeepPot-SE | | DPA-1 | | ASDP (ours) | |
|---|---|---|---|---|---|---|---|---|---|---|
| | ΔE (meV) | ΔF (meV/Å) | ΔE (meV) | ΔF (meV/Å) | ΔE (meV) | ΔF (meV/Å) | ΔE (meV) | ΔF (meV/Å) | ΔE (meV) | ΔF (meV/Å) |
| AlMgCu | 226.4 | **60.8** | N/A | N/A | 62.0 | 86.0 | **9.3** | 68.8 | 19.5 | 63.9 |
| ANI-1 (large) | N/A | N/A | N/A | N/A | **15.8** | 199.0 | 16.8 | 193.0 | 24.5 | **185.0** |
| SSE-PBE | 434.1 | **61.9** | N/A | N/A | 7.7 | 94.0 | 3.8 | 103.0 | **2.3** | 94.8 |
| HECN | 557.6 | 271.8 | N/A | N/A | **11.1** | 279.0 | 12.5 | 265.0 | 11.6 | **253.0** |
| 2D-In$_2$Se$_3$ | 211.8 | **98.9** | 787.0 | 173.7 | **14.0** | 136.0 | 15.8 | 149.0 | **14.0** | 134.0 |
| Organic-reaction | N/A | N/A | N/A | N/A | 74.7 | 219.0 | 76.1 | 224.0 | **63.3** | **199.0** |

Table 1: Root Mean Square Errors (RMSE) for energy (meV) and force (meV/Å) predictions on the 6 selected molecular system datasets, comparing our proposed model, ASDP, against several SOTA baselines. The lowest error values for each task are highlighted in **bold**, the second lowest values are underlined, and N/A denotes the training of the model is **not converged**.

| Systems | $s = 0$ | | $s = N_1$ | | $s = N_2$ | | $s = 30$ | |
|---|---|---|---|---|---|---|---|---|
| | ΔE (meV) | ΔF (meV/Å) | ΔE (meV) | ΔF (meV/Å) | ΔE (meV) | ΔF (meV/Å) | ΔE (meV) | ΔF (meV/Å) |
| AlMgCu | 45.2 | 80.3 | 27.0 | 69.7 | **19.5** | **63.9** | 21.4 | 68.6 |
| ANI-1 (large) | **20.3** | 188.0 | 30.3 | 187.0 | 24.5 | **185.0** | 27.5 | 187.0 |
| SSE-PBE | 4.2 | 97.0 | 3.0 | 96.1 | **2.3** | **94.8** | 4.1 | 95.1 |
| HECN | 12.4 | 261.0 | 17.9 | 264.0 | **11.6** | **253.0** | 14.3 | 261.0 |
| 2D-In$_2$Se$_3$ | 15.2 | 140.0 | **13.9** | 135.0 | 14.0 | **134.0** | 15.5 | 137.0 |
| Organic-reaction | 90.4 | 204.0 | 65.5 | **199.0** | **63.3** | **199.0** | 81.4 | 201.0 |

Table 2: Ablation study validating the choice of the shell-aware hyperparameter $s$ for ASDP. This table displays the [Energy/Force] RMSE for four settings of $s$: a radial-only model ($s = 0$), using the first shell ($s = N_1$), using the first two shells ($s = N_2$), and a large cutoff ($s = 30$).

improvement for modeling ion diffusion barriers. Similarly, it achieves the lowest energy (63.3 meV) and force (199 meV/Å) errors for the organic-reaction dataset, showcasing its ability to accurately map complex potential energy surfaces. This consistent superiority in force prediction is especially significant. In molecular dynamics, force accuracy is more critical than energy because forces directly determine atomic accelerations and propagate trajectories via Newton's equations of motion. Their fidelity is therefore paramount for ensuring a simulation's stability and physical realism. In systems like AlMgCu and ANI-1 (large), where even if DPA-1 shows a slightly lower energy RMSE, ASDP consistently yields more accurate forces (e.g., 63.9 vs. 68.8 meV/Åfor AlMgCu). This demonstrates that while DPA-1 may suffice to approximate global energy, ASDP's physically-grounded mechanism captures forces with higher fidelity. This advantage is further confirmed by its leading force accuracy on the HECN system.

## 4.4 ABLATION STUDIES

We performed an ablation study to demonstrate that confining angular information to the first two coordination shells is an effective physical prior for ASDP. We systematically varied the shell-aware hyperparameter, $s$, testing four configurations for each system: (i) a radial-only baseline ($s = 0$); (ii) the first coordination shell only ($s = N_1$); (iii) the first and second shells ($s = N_2$, our proposed model); and (iv) a large, non-physical cutoff ($s = 30$), and all other training parameters were held constant. Note that the shell boundaries, $N_1$ and $N_2$, are fixed constants for all atoms within a given system, determined from a priori chemical knowledge (see Appendix A.2 for detailed settings).

The results in Table 2 validate that confining angular information to the first two shells ($s = N_2$) is an optimal strategy. The necessity of angular features is demonstrated by the purely radial model ($s = 0$), where performance degrades significantly (e.g., AlMgCu energy error rises from 19.5 to 45.2 meV). Conversely, an excessive cutoff ($s = 30$) introduces noise and also harms accuracy, as seen in the Organic-reaction system. The $s = N_2$ configuration consistently achieves the best or most competitive accuracy, particularly for forces, which are critical for dynamics. Even when another setting yields a marginally better energy (e.g., ANI-1), $s = N_2$ provides superior force

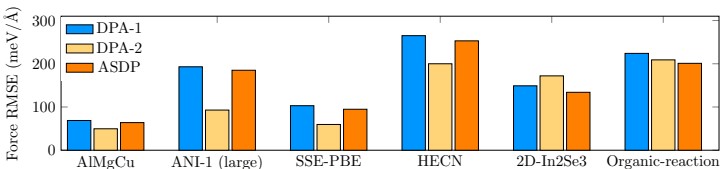

Figure 4: Comparison of Force Root Mean Square Error (RMSE, in meV/Å) for DPA-1, DPA-2, and ASDP models across six benchmark systems.

predictions (185.0 meV/Å). This confirms that $s = N_2$ acts as a robust physical prior, effectively balancing descriptive power and model focus.

### 4.5 ANALYSIS AND DISCUSSION

**Synergy with Large Atomic Models** A key trend in deep potentials is the rise of Large Atomic Models (LAMs) like DPA-2 Wang (2024), which seek broad generalization through massive parameter counts, contrasting with specialized, lightweight models like DPA-1 and ASDP. Adopting the official default hyperparameters for our analysis, the difference in scale between these models becomes pronounced: DPA-2 features 1.79M parameters, nearly three times the 613.2k of DPA-1 and 614.2k of ASDP. Figure 4 directly illustrates the resulting performance trade-off from this disparity in scale. The large DPA-2 model excels on diverse systems like AlMgCu, ANI-1, and SSE-PBE, showcasing the power of scale for broad applicability, albeit at a higher computational cost. Conversely, the lightweight models provide an efficient alternative, delivering competitive or even slightly higher accuracy on specialized tasks such as 2D-In$_2$Se$_3$ and Organic-reaction. More importantly, **the shell-aware angular encoding introduced in ASDP is a validated and portable methodology**. Its principles can be integrated into descriptor-based LAMs like DPA-2 to potentially boost their descriptive power. Therefore, a key direction for our future work is to integrate the mechanism of our shell-aware angular encoding into the architecture of LAMs, aiming to create models that are both broadly generalizable and highly precise.

**Limitations and Future Directions for Shell-Aware Encoding** The current ASDP implementation employs a static hyperparameter, $s$, to define the number of neighbors for angular encoding, empirically approximating the count for the first two coordination shells across an entire system. While this approach has validated the core concept of shell-aware angular encoding, it does not capture the dynamic nature of local coordination, which varies per-atom and over time. A crucial direction for future work is therefore the development of a dynamic and adaptive shell-determination strategy. Such a method would allow the model to ascertain the ideal neighbor count on-the-fly for each atom, enabling the descriptor to adapt to the precise, instantaneous local environment and potentially yielding further gains in accuracy and robustness.

## 5 CONCLUSION

In this work, we introduced the Angular and Shell-Aware Deep Potential (ASDP), an architecture that addresses key limitations in attention-based models by integrating a learnable, shell-constrained angular bias. This physically-motivated prior focuses the model on chemically-significant interactions within the first two coordination shells, effectively filtering angular noise from distant atoms. Across a diverse benchmark, ASDP delivered state-of-the-art or highly competitive performance, with consistently superior force predictions that are critical for stable and high-fidelity molecular dynamics simulations. The success of ASDP validates the embedding of physical priors within a learnable descriptor as a robust design principle for next-generation machine learning potentials, enabling more accurate and efficient large-scale atomistic modeling.

### DECLARATION OF LLM USAGE

The usage of LLMs is strictly limited to aid and polish the paper writing.

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

# A APPENDIX

## A.1 ADDITIONAL RELATED WORK

**Descriptor-based PES Model.** These methods operate on a two-stage principle: first, a descriptor module encodes the local atomic environment into a fixed-size, symmetry-invariant feature vector, which is then mapped to an atomic energy by a fitting network. Initial works relied on hand-crafted descriptors with a fixed functional form, leading to a proliferation of powerful representations including those found in BPNN Behler & Parrinello (2007), GAP Bartók et al. (2010), SNAP Thompson et al. (2015), MTP Shapeev (2016), ANI Smith et al. (2017), TensorMol Yao et al. (2018), ACE Drautz (2019) and others Unke & Meuwly (2018); Artrith et al. (2017); Khorshidi & Peterson (2016). A conceptual leap was the introduction of learnable descriptors in the Deep Potential (DP) model Han et al. (2017); Zhang et al. (2018a;b), where a network learns the representation itself. This direction has recently culminated in the Deep Potential-Attention (DPA-1) model Zhang et al. (2024). By integrating an attention mechanism to dynamically weight the importance of neighboring atoms, DPA-1 significantly boosts the descriptor's expressive power. This advancement, combined with the inherent computational efficiency and parallelism-friendly nature of the descriptor-based framework, has established it as a state-of-the-art approach that offers a formidable balance of accuracy and speed for large-scale simulations.

**Message-Passing PES Model.** The evolution of message-passing potentials began with foundational invariant models like SchNet Schütt et al. (2018), CGCNN Xie & Grossman (2018), and PhysNet Unke & Meuwly (2019), which learned interactions from scalar distance features. A subsequent wave of architectures sought to encode richer geometric information; models like DimeNet/DimeNet++ Gasteiger et al. (2020b;a), SphereNet Liu et al. (2022), and GemNet Gasteiger et al. (2021) introduced directional or specialized messages, while others like OrbNet Qiao et al. (2020) explored deeper interaction models and explicit quantum features. Influenced by works like SE(3)-Transformers Fuchs et al. (2020), a major conceptual shift arrived with E(3) equivariant networks like PaiNN Schütt et al. (2021) and NequIP Batzner et al. (2022), which preserve complete geometric information by using features that co-transform with the input coordinates. However, a common limitation uniting these approaches is their inherent difficulty in parallelization, as each message-passing step depends on the completion of the previous one. To overcome this scalability challenge, the latest generation of models like Allegro Musaelian et al. (2023) and MACE Batatia et al. (2022) enforces strict locality, which enables dramatic improvements in parallelism and linear scaling but introduces the trade-off of increased computational complexity for each local interaction.

## A.2 DETAILS FOR THE MOLECULAR SYSTEM DATASETS

This sub-section provides a detailed description of the six datasets used to benchmark our model, ASDP, against existing methods. For each system, we outline its origin, the strategy for train/test splitting, and the rationale behind the chosen shell configurations ($N_1$ and $N_2$), where $N_1$ denotes the number of neighbors in the first coordination shell and $N_2$ is the total neighbor count up to the

| System | Atom Types | Dataset Size | | Shell Config | |
|---|---|---|---|---|---|
| | | # Train Frames | # Test Frames | $N_1$ | $N_2$ |
| AlMgCu | Al, Mg, Cu | 86,322 | 5,790 | 12 | 18 |
| ANI-1 (large) | C, H, N, O | 35,954 | 1,858 | 4 | 12 |
| SSE-PBE | Li, P, S, Cl | 15019 | 755 | 6 | 18 |
| HECN | C, N, Ti, Zr, Hf, Ta, Nb | 7,351 | 2,547 | 6 | 18 |
| 2D-In2Se3 | In, Se | 7,797 | 1,543 | 6 | 12 |
| Organic-reaction | C, H, N, O | 19,318 | 5,931 | 4 | 12 |

Table 3: A summary of the six molecular system datasets used for benchmarking. This includes the constituent atom types, the number of frames in the training and testing sets, and the shell neighbor configurations ($N_1$ and $N_2$) determined from *a priori* chemical knowledge. $N_1$ denotes the number of neighbors in the first coordination shell, while $N_2$ is the total count up to the second shell (which equals to the $s$ used in ASDP).

second shell. It is important to note that in our ASDP implementation, the second-shell neighbor count is treated as a fixed hyperparameter, denoted as $s$, for each system, where we set $s = N_2$. This means all atoms within all frames of a given dataset are assigned this same universal s value, a simplification that greatly facilitates efficient parallelization of the computation. A summary of key statistics is provided in Table 3.

Moreover, the datasets for AlMgCu, SSE-PBE, HECN, 2D-In$_2$Se$_3$, and Organic-reaction are sourced from AIS Square[1], while ANI-1 is from GitHub; direct links and detailed train/test splits for all systems are available in our open-source repository[2].

### A.2.1 ALMGCU SYSTEM

**Origin** The AlMgCu system dataset was generated via the DP-GEN Zhang et al. (2020) concurrent learning scheme, which explored 2.73 billion alloy configurations to produce a compact, labeled dataset of 100,000 structures. The dataset is significant for its comprehensive coverage of the entire Al-Mg-Cu compositional space (including single, binary, and ternary systems) and an extremely wide range of thermodynamic conditions (50.0 K to 2579.8 K and 1 bar to 50,000 bar). This diversity makes it a robust and challenging benchmark for testing interatomic potentials. The dataset was originally presented by Jiang et al. (2021).

**Train/Test Splitting** Following the experimental setup of DPA-1 Zhang et al. (2024), we partitioned the AlMgCu dataset to rigorously evaluate the model's transferability. We composed the training set from all **unary** and **binary** alloy configurations, resulting in 86,322 frames. The test set was exclusively formed from the more complex ternary systems, comprising the remaining 5,790 frames. This strategy is specifically designed to assess the model's ability to extrapolate from simpler, constituent systems to predict the properties of more complex alloys.

**Shell Configuration** The neighbor shell configuration for the AlMgCu system is determined by the inherent crystal structures of its constituent elements. Specifically, Aluminum (Al) and Copper (Cu) typically form a Face-Centered Cubic (FCC) lattice, while Magnesium (Mg) adopts a Hexagonal Close-Packed (HCP) structure. Both FCC and HCP are close-packed arrangements, and based on established crystallographic knowledge for these structures, an atom has 12 nearest neighbors, which constitute the first coordination shell ($N_1$). The second coordination shell contains the next 6 nearest neighbors. Therefore, the number of neighbors up to the second shell ($N_2$) is set to 18.

### A.2.2 ANI-1 (LARGE) SYSTEM

**Origin** This is a subset of the extensive ANI-1 dataset Smith et al. (2017), focusing on organic molecules containing C, H, N, and O with the atom number larger than 40. It is a standard benchmark for machine learning potentials, testing the ability to model covalent bonding in diverse chemical environments.

**Train/Test Splitting** We adopted the official train/test partition for the ANI-1 (large) dataset. This resulted in a training set containing 35,954 frames and a test set of 1,858 frames.

**Shell Configuration** The fixed atomic neighborhood for the ANI-1 dataset, defined by a total of $N_2 = 12$ atoms partitioned into a $4 + 8$ two-shell structure, is a deliberate strategy designed to balance descriptive accuracy with computational efficiency. The first shell of $N_1 = 4$ is specifically chosen to accommodate the tetravalency of carbon, the foundational element of organic chemistry, ensuring that the immediate covalent bonding environment of any atom is fully captured, as this value serves as a robust upper bound for all relevant elements in the dataset. Subsequently, the second shell of 8 atoms acts as a pragmatic approximation for the more variable population of next-nearest neighbors; this count is sufficient to encode crucial non-local effects like steric hindrance, which significantly influences molecular conformation and energy, for a vast majority of typical structures. This complete $4 + 8$ configuration, resulting in the $N_2 = 12$ total, thus creates a chemically meaningful, fixed-size descriptor that provides an accurate representation of core bonding while also supplying the necessary context for key non-bonded interactions.

---

[1] https://www.aissquare.com/
[2] https://anonymous.4open.science/r/ASDP-ICLR-code

### A.2.3 SSE-PBE SYSTEM

**Origin** The SSE-PBE system Huang et al. (2021) originates from a class of $Li_{10}XP_2S_{12}$-type solid-state electrolyte (SSE) materials. In these materials, the 'X' site is occupied by either a single element or a combination of elements from the group of Germanium (Ge), Silicon (Si), and Tin (Sn). The system encompasses both ordered structures, where atomic positions are fixed, and disordered structures, where specific sites are randomly occupied by Ge, Si, Sn, or P atoms.

**Train/Test Splitting** We adopted the official train/test partition for the SSE-PBE dataset. This resulted in a training set containing 15,019 frames and a test set of 755 frames.

**Shell Configuration** The local atomic environment is described by a two-shell model where the first coordination shell ($N_1$) includes 6 neighbors and the total neighbor count ($N_2$) is 18. This $N_1 = 6$ cutoff is deliberately chosen to be comprehensive: it not only includes the stable tetrahedral coordination (CN=4) of the Ge/Si/Sn and P framework atoms with sulfur but also fully captures the more variable, higher-coordination environments (up to CN=6) of mobile Li ions and bridging S atoms. The extension to $N_2 = 18$ is essential for modeling the medium-range order that dictates the material's function, capturing critical interactions such as the arrangement of neighboring Li ions, which governs ionic conductivity, and the connectivity between adjacent $[XS_4]$ and $[PS_4]$ tetrahedra, which controls overall structural stability.

### A.2.4 HECN SYSTEM

**Origin** The HECN system dataset Baidyshev et al. (2024); Nikitin et al. (2025) is a collection of structural configurations for high-entropy carbides (HEC) and high-entropy carbonitrides (HECN), based on the materials studied in references. This dataset is comprehensive, containing both ordered crystal configurations and disordered amorphous configurations. The amorphous structures specifically correspond to the liquid phases of both HEC and HECN, providing a basis for modeling these materials across different states of matter.

**Train/Test Splitting** The dataset was divided into training and test sets using an 80:20 ratio applied at the system level to prevent data leakage. This system-wise split resulted in a training set containing 7,351 frames and a test set containing 2,547 frames.

**Shell Configuration** The local atomic environment is described using a two-shell model, with the first coordination shell ($N_1$) set to 6 and the total number of neighbors ($N_2$) set to 18. This configuration is directly derived from the fundamental rock-salt (NaCl-type) crystal structure characteristic of high-entropy carbides and carbonitrides (HEC/HECN). In this highly symmetric lattice, every atom is perfectly and octahedrally coordinated by exactly 6 atoms of the other type, making $N_1$=6 a physically precise choice for the first shell. Specifically, each metal atom is surrounded by 6 C/N atoms, and each C/N atom is surrounded by 6 metal atoms. The second coordination shell in this structure consists of the next 12 nearest neighbors, which are atoms belonging to the same sublattice as the central atom. Therefore, the total neighbor count of $N_2 = 18$ (6 first-shell + 12 second-shell) provides a complete and accurate description of the first two coordination spheres.

### A.2.5 2D-$In_2Se_3$ SYSTEM

**Origin** This dataset models the 2D material Indium Selenide ($In_2Se_3$), which is known for its ferroelectric properties.

**Train/Test Splitting** The dataset was divided into training and test sets using an 80:20 ratio applied at the system level to prevent data leakage. This system-wise split resulted in a training set containing 7797 frames and a test set containing 1543 frames.

**Shell Configuration** The shell configuration was defined with a first coordination shell of $N_2 = 6$ and a total neighbor count of $N_2 = 18$, a physically-informed choice designed to accommodate the rich structural polymorphism of 2D-$In_2Se_3$. The selection of $N_1 = 6$ serves as an inclusive upper bound, robustly capturing the varied first-shell coordination numbers (CN) present across the material's phases, which range from 3 (for outer Se atoms) to 4 (tetrahedral In) and 6 (central Se or octahedral In). Furthermore, the addition of 12 neighbors for the second shell is a structurally-motivated decision based on the specific atomic arrangement within the material's quintuple-layer structure. For any given atom, this second shell is primarily composed of two distinct groups: ap-

proximately 6 in-plane neighbors within the same hexagonal-like sublattice (e.g., In-In interactions) and additional cross-layer atoms from adjacent layers. This choice ensures that the model captures essential medium-range interactions, encompassing both in-plane metallic character and cross-layer covalent/ionic influences. Therefore, the total neighbor count of $N_2 = 18$ provides a comprehensive descriptor that accounts for both the immediate bonding and the complex medium-range order critical to defining the properties of 2D-$In_2Se_3$.

### A.2.6 ORGANIC-REACTION SYSTEM

**Origin** This dataset was obtained from the Transition-1x dataset Schreiner et al. (2022) and designed to test the model's ability to describe chemical reactivity, containing snapshots along a reaction coordinate for an organic reaction involving C, H, N and O.

**Train/Test Splitting** The dataset was divided into training and test sets using an 80:20 ratio applied at the system level to prevent data leakage. This system-wise split resulted in a training set containing 19,318 frames and a test set containing 5,931 frames.

**Shell Configuration** Given the strong similarity in molecular composition between the organic-reaction dataset and the ANI-1 (large) dataset, we adopted its shell configuration. We therefore set the first coordination shell to $N_1 = 4$ and the total neighbor count to $N_2 = 12$.

### A.3 EXPERIMENTAL SETUP

### A.3.1 MODEL SETTINGS

All descriptor-based models, namely DeepPot-SE, DPA-1, DPA-2, and ASDP, were implemented using the Deepmd-kit framework [3]. These models share a common fitting network architecture: a three-layer Multi-Layer Perceptron (MLP) with 240 neurons in each layer. The descriptor architecture for these models begins with a shared embedding network, which is a three-layer MLP with 25, 50, and 100 neurons in the respective layers. Building upon this, the DPA-1 and ASDP descriptors incorporate a self-attention module composed of two attention layers, each with an output dimension of 128. In contrast, the DPA-2 descriptor is constructed from a reprint layer followed by a repformer layer, with its hyperparameters configured according to the default settings in the official documentation. A universal cutoff radius (rcut) of 6.0 Åwas applied across all models. The number of selected neighbors (nsel) was tailored to each system by ensuring the value was greater than or equal to the maximum number of neighbors any atom had within the cutoff radius. Accordingly, the nsel values were set to 140 for AlMgCu, 60 for ANI-1 (large), 85 for HECN, 40 for 2D-$In_2Se_3$, 15 for organic-reaction, and 60 for SSE-PBE.

For the remaining two message-passing-based baseline models, namely, Nequip[4] and Allegro[5], they are configured with a consistent set of foundational hyper-parameters to ensure a fair comparison. Both models employed a cutoff radius (r_max) of 5.0 Å, a maximum spherical harmonic order (l_max) of 1, and enabled parity (parity=true). Furthermore, a Ziegler-Biersack-Littmark (ZBL) pair potential was integrated into both models as a physical prior for short-range repulsion.

The specific architectural configurations for each model are detailed below:

- Nequip was configured with 4 interaction blocks (num_layers) and a feature multiplicity of 32 (num_features). The radial basis was defined by 8 Bessel functions (num_bessels), and the internal radial MLP network consisted of 2 layers (radial_mlp_depth) with a width of 64 neurons (radial_mlp_width).

- Allegro was constructed with 2 interaction layers (num_layers). It was defined with a scalar feature dimension of 64 (num_scalar_features) and a tensor feature dimension of 32 (num_tensor_features). A key characteristic of this configuration is that all internal multi-layer perceptrons (MLPs)—including the scalar embedding, interaction, and readout networks—were uniformly structured with a depth of 1 layer and a width of 64 neurons,

---

[3]`https://github.com/deepmodeling/deepmd-kit`
[4]`https://github.com/mir-group/nequip`
[5]`https://github.com/mir-group/allegro`

using the SiLU activation function. The radial embedding utilized the same 8-function Bessel basis as the Nequip model.

### A.3.2 TRAINING DETAILS

All models (including ASDP and the baselines) are trained and evaluated using one NVIDIA A100 80GB GPU. Models including DeepPot-SE, DPA-1, DPA-2 and ASDP are trained by finding the model parameters $w$ that minimize a weighted Mean Squared Error (MSE) loss, $\mathcal{L}(w)$, averaged over a mini-batch $\mathcal{B}$. The objective function is defined as:

$$\mathcal{L}(w) = \frac{1}{|\mathcal{B}|} \sum_{l \in \mathcal{B}} \left( p_\epsilon |E_l - E_l^w|^2 + p_f |\mathcal{F}_l - \mathcal{F}_l^w|^2 \right) \tag{12}$$

In this equation, $E_l$ and $\mathcal{F}_l$ represent the ground-truth energy and forces for a given sample $l$. The terms $E_l^w$ and $\mathcal{F}_l^w$ are the corresponding values predicted by the model, which is explicitly parameterized by its weights $w$. The coefficients $p\epsilon$ and $p_f$ are hyperparameters that balance the relative importance of the energy and force terms in the total loss. In our experiments, the start learning rate and stop learning rate are set to 1e-3 and 3.51e-6, with the decay steps of 5000. During training, $p\epsilon$ is increased from 0.02 to 1, while $p_f$ is decayed from 1000 to 1, following the same experimental setting as DPA-1's official implementation. The training protocol was standardized across all systems to ensure robust convergence and optimal model selection. For each system, the batch size was individually tuned to fully utilize the memory of a single GPU. All models were trained for a total of 100,000 steps, a duration determined to be sufficient for convergence. An exception was the AlMgCu system, which required an extended training of 200,000 steps owing to a slower convergence rate. Throughout the training process, model performance was evaluated every 1,000 steps. The final model was then selected by identifying the checkpoint that corresponded to the minimum force Root Mean Square Error (RMSE) observed across all evaluations.

The training protocols for both the NequIP and Allegro models shared several key details. A balanced loss function was employed, with the weight coefficients for both total energy and forces set to 1.0. Both models were trained using the Adam optimizer with an initial learning rate of 0.01. Consistent with the descriptor-based models, the batch size for each system was individually tuned to fully utilize the memory of a single GPU. For the NequIP model specifically, an Exponential Moving Average (EMA) of the weights was applied with a decay factor of 0.999. To ensure convergence, a large maximum number of epochs (e.g., 1500) was set for each training run. However, the training process was actively monitored by evaluating the model on a validation set after each epoch. The training was truncated once performance converged, and the checkpoint that yielded the minimum force Root Mean Square Error (RMSE) was selected as the final model.

### A.4 ADDITIONAL EXPERIMENTAL RESULTS

### A.4.1 ABLATION STUDY OF ANGULAR ENCODING FEATURES

As detailed in Subsection 3.3 and Equation 9, the angular encoding module is designed to generate six features for each angle $\theta_{j,k}^i$. These features are: $\cos(\theta_{j,k}^i)$, $\sin(\theta_{j,k}^i)$, $\sin(2 \cdot \theta_{j,k}^i)$, $f_{\mathrm{vm}}(\theta_{j,k}^i \mid \kappa)$, $\mathbf{r}_{ij} + \mathbf{r}_{ik}$ and $|\mathbf{r}_{ij} - \mathbf{r}_{ik}|$. The first 4 features capture purely angular information, while the latter 2 provide length information associated with the vectors forming the angle, which can represent bond length information within the first coordination shell. To investigate the contribution of this explicit bond length information, we designed an ablation study. We compared two model configurations: one using only the first four angular features (denoted as "ASDP w. 4 features") and another using the complete set of six features (denoted as "ASDP w. 6 features") for both training and inference.

The performance of these two model configurations across various systems, as presented in Table 4, reveals that the inclusion of bond length features does not lead to a universally superior model, and its impact is system-dependent. For instance, the "ASDP w. 6 features" model shows a clear advantage for the HECN and Organic-reaction systems, significantly reducing the energy error and also improving the force error. This suggests that for these chemically complex systems, the explicit encoding of bond length information is crucial for accurately representing the potential energy surface. Conversely, for the ANI-1 (large) and 2D-$In_2Se_3$ systems, the "ASDP w. 4 features" model

| Systems | ASDP w. 4 features | | ASDP w. 6 features | |
|---|---|---|---|---|
| | $\Delta E$ (meV) | $\Delta F$ (meV/Å) | $\Delta E$ (meV) | $\Delta F$ (meV/Å) |
| AlMgCu | **18.5** | 64.0 | 19.5 | **63.9** |
| ANI-1 (large) | **16.3** | 186.0 | 24.5 | **185.0** |
| SSE-PBE | 2.5 | 95.0 | **2.3** | **94.8** |
| HECN | 12.8 | 259.0 | **11.6** | **253.0** |
| 2D-$In_2Se_3$ | **11.0** | **131.0** | 14.0 | 134.0 |
| Organic-reaction | 89.8 | 202.0 | **63.3** | **199.0** |

Table 4: Performance comparison of the ASDP model with 4 and 6 angular encoding features. The table displays the energy RMSE ($\Delta E$) in meV and force error ($\Delta F$) in meV/Åacross various systems. The best-performing result for each metric is shown in **bold**.

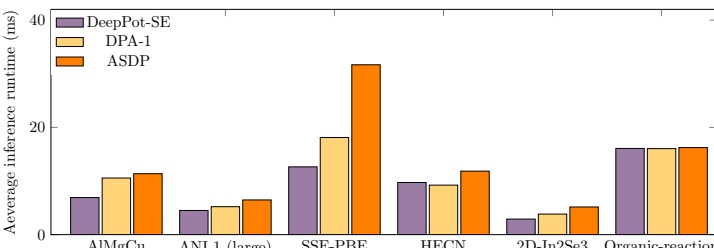

Figure 5: Comparison of Average Inference Runtimes for ASDP, DPA-1, and DeepPot-SE Models.

achieves lower errors in both energy and force, indicating that the additional length features might introduce noise or redundancy in these contexts. Furthermore, in the AlMgCu and SSE-PBE systems, the performance difference between the two models is minimal, with nearly identical force errors, suggesting that the length information has a negligible effect on force prediction in these cases.

We conclude that the utility of incorporating explicit bond length information into the angular encoding module is highly system-dependent. While it can be beneficial for certain systems, particularly those with complex organic or covalent interactions like HECN and Organic-reaction, it may not be universally advantageous and can even slightly degrade performance in other cases.

### A.4.2 INFERENCE RUNTIME ANALYSIS

To validate the computational efficiency of ASDP for molecular dynamics (MD) applications, we benchmarked its average inference runtime against DeepPot-SE and DPA-1 models. For this benchmark, we measured the time required to compute the energy and forces for each individual frame within the test set of every system. All tests were conducted on a single NVIDIA A100 GPU to ensure a fair comparison, and the final values reported in Figure 5 represent the average of these per-frame timings. The analysis shows that ASDP maintains competitive efficiency, ensuring its practicality for simulations. On the majority of systems, including AlMgCu, HECN, ANI-1 (large), and 2D-$In_2Se_3$, ASDP's inference time is only marginally higher than the baselines. Notably, for the Organic-reaction system, all three models exhibit nearly identical runtimes, demonstrating high comparability. While a higher computational cost is observed for ASDP on the SSE-PBE system, its performance across the diverse range of other systems confirms that its efficiency is largely on par with state-of-the-art models. This ensures that ASDP can be deployed effectively in demanding MD simulations without introducing significant computational overhead.

