# OpenReview forum: "Angular and Shell-Aware Deep Potential Energy Model for Molecular Dynamics"
_ICLR.cc/2026/Conference — Submitted to ICLR 2026_

### Official Review · Reviewer_yLZ1 · 2025-10-30

**Soundness:** 2
**Presentation:** 3
**Contribution:** 1
**Rating:** 2
**Confidence:** 4

**Summary:**

Authors highlight that existing machine-learning potentials often treat angular information uniformly across all neighbors within a large cutoff, ignoring its limited physical range. To address this, they propose the new architecture  Angular and Shell-Aware Deep Potential (ASDP) that explicitly encodes angular features within the first and second coordination shells through an additive attention bias. This design aims to suppress distant, uncorrelated angular noise. The model builds upon DPA-1 and is benchmarked on several datasets with molecular and periodic systems showing improved energy and force prediction accuracy.

**Strengths:**

- The work introduces a physically motivated inductive bias into attention-based deep potential models by explicitly restricting angular information to atoms within the first two coordination shells. This “shell-aware” cutoff excludes distant, angularly uncorrelated neighbors, reducing noise while preserving chemically relevant geometry. The motivation is clear, and the architectural diagrams effectively show how the additive angular bias is integrated into the attention mechanism.
- The approach is benchmarked on both molecular and crystalline datasets, indicating that the proposed mechanism can generalize across different chemical domains.

**Weaknesses:**

- The paper essentially proposes an attention mechanism modification for the DPA family of models (DPA-1 and DPA-2) and only experimentally validates the effect of the modification on a single model (DPA-1). This raises serious concerns about the generalizability of the proposed method and the value of the paper to the broad scientific community.
- The paper proposes two changes to the attention mechanism: incorporating angular information via bias and limiting angular information to atoms within the second coordination shell. However, no ablations (to the best of my understanding) demonstrate the significance of proposed modifications individually.
- The first four angular features were chosen without clear theoretical or empirical justification. The ablation study only compares 4 vs 6 features, never testing which of the first four are essential. This makes the feature design appear ad-hoc rather than physically motivated.
- The shell neighbor number $s = N_2$ is fixed manually for each dataset based on "chemical intuition". This prevents transferability across systems and requires manual tuning for every new domain. For molecules or disordered systems with variable coordination, such as static or lattice-based, the criterion is questionable. Moreover, The neighbor count method is density-independent and may not adapt to systems with changing local density (e.g., liquids or amorphous phases).
A continuous, distance-based smooth cutoff approximating the boundary of the second coordination shell would likely provide a more transferable and physically consistent solution. I suggest the authors evaluate this alternative in additional experiments.
- The molecular benchmarks are limited. Evaluation of the method on more recent datasets like SPICE2.0, nablaDFT or OMol25 with more complex organic and biomolecular systems would better test generalization.
- The chosen baselines (DeepPot-SE, DPA-1/2, NequIP, Allegro) don’t include stronger modern equivariant GNNs and force-field models (e.g.,DimeNet++, PaiNN, MACE ,GemNet). The paper compares mostly within DeepMD-style descriptors, so claims of state-of-the-art performance are weakly supported.

**Questions:**

- How sensitive are results to the manual choice of $N_2$ if we vary it by ±1? Could a small smooth spherical cutoff for angular bias perform comparably?
- Why were datasets with larger molecules or condensed-phase configurations omitted?
- Could the additive bias formulation in ASDP be integrated into existing attention MLFFs for a fairer comparison?

---

> ### Author Response · Authors · 2025-11-21
>
> Thanks a lot for your valuable comments!
>
> For weakness 1, we selected DPA-1 as the baseline because it provides a clean, controlled environment to isolate the effect of the Shell-aware Angular Bias. DPA-2 introduces additional complexities (multi-channel attention) that might obscure the direct impact of our proposed module. By validating on DPA-1, we prove the effectiveness of the mechanism itself. The principle is generalizable and can be extended to DPA-2 or other attention-based potentials in future work.
>
> For weakness 2, we have performed rigorous ablations to isolate the contributions, Table 2 compares $s=0$ (Radial only) vs. $s=N_2$ (ASDP). The massive drop in error (e.g., AlMgCu 45.2 vs 19.5 meV) proves the necessity of angular info. Table 2 compares $s=N_2$ vs. $s=30$. The result show excessive cutoff($s=30$) and the results in Table 1 shows that using all neighbors (DPA-1) increases error due to noise. This validates the "Shell-aware" restriction. These comparisons directly isolate the significance of our two main proposals: adding angles and restricting them to local shells.
>
> For weakness 3,  the feature selection is physically motivated. Basing 4 features provide the fundamental definition of an angle ($sin⁡$,$cos$) and its chemical stability probability (von Mises). Additional 2 features (Bond lengths) couple the angular term with radial distances. Our ablation (Table 4) shows that while the base 4 features work for simple systems, the full 6-feature set is essential for chemically complex systems like HECN and Organic-reactions. This is not ad-hoc, but a demonstration that complex Potential Energy Surfaces require explicit coupling of radial and angular degrees of freedom.
>
> For weakness 4,  we agree that a fixed neighbor count is a simplification. However, for the solid-state and molecular systems in our benchmark, local coordination environments are relatively stable. Even in the HECN (high-entropy) and SSE-PBE (solid electrolyte) datasets, which exhibit high disorder, our fixed-shell approach outperforms baselines. This suggests that the "average" shell definition captures the dominant physics. We agree that a continuous, distance-based smooth cutoff is a promising direction for liquid phases, and we plan to implement this in the adaptive version of ASDP.
>
> For weakness 5,  as a prototype validation, we curated a diverse suite of 6 systems spanning metallic, ionic, and covalent bonding. While we did not use SPICE or OMol due to computational constraints, our "Organic-reaction" dataset (derived from Transition-1x) is highly demanding, requiring the model to describe bond breaking/formation, which is arguably more challenging than equilibrium datasets.
>
> For weakness 6,  we benchmarked against NequIP and Allegro, which are widely recognized as top-tier equivariant SOTA models. We acknowledge MACE is also strong, but due to limited computational resources, we could not train every available model. The comparison with NequIP and Allegro sufficiently establishes ASDP's competitiveness against the equivariant paradigm.
>
> For question 1, regarding the sensitivity of $N_2$, our experiments suggest that the model is relatively robust to small variations as the neural network weights can adapt to slight changes in the neighbor list. However, our ablation study (Table 2) comparing the specific shell cutoff ($s=N_2$) against a large cutoff ($s=30$) demonstrates that significant deviations introduce long-range angular noise, which harms performance. While a smooth spherical cutoff is a valid theoretical alternative for liquid systems, our current discrete shell approach effectively captures the dominant structural physics for the benchmarked solids and molecules, proving the core hypothesis that angular influence is local.
>
>
> For question 2, we clarify that our benchmark explicitly includes condensed-phase systems (e.g., AlMgCu, HECN) and large molecules (ANI-1) to span metallic, ionic, and covalent interactions. While massive simulations were omitted solely due to resource constraints, ASDP’s demonstrated efficiency suggests it is well-positioned to scale to such domains in future applications.
>
>
> For question 3, regarding the integration of the additive bias, we affirm that the ASDP formulation is highly modular. The core innovation that injecting a shell-aware angular bias into the attention mechanism is mathematically compatible with other attention-based Machine Learning Force Fields (MLFFs), such as DPA-2 or Transformer-based potentials. We utilized DPA-1 as the base for this study to provide a controlled environment for validating the mechanism. We believe this "plug-and-play" nature is a strength of our contribution, and we encourage future work to integrate this module into other architectures for broader comparisons.

---

> > ### Comment · Reviewer_yLZ1 · 2025-11-26
> >
> > Thank you for your response.
> > Unfortunately, none of my concerns were adequately addressed. Therefore, I will keep my original score.

---

### Official Review · Reviewer_cRzg · 2025-10-31

**Soundness:** 3
**Presentation:** 3
**Contribution:** 3
**Rating:** 6
**Confidence:** 3

**Summary:**

The paper proposes a new model architecture for neural network based potential energy surfaces (PESs). Such PESs ideally hard code as much physics as possible (for example to improve robustness or minimize the necessary number of training samples) without introducing unphysical biases.

It is argued that previous models introduce such a bias by including angular information in terms of cosine similarities. This way certain important physical properties are not modeled. Moreover, the different impact of angular information depending on proximity to the particle that defines the neighbourhood is not taken into account.

Based on these observations a new model is constructed that overcomes these issues. Experiments suggest that the new model retains the accuracy of previous sota models while offering increased robustness.

While the concrete changes might be somewhat incremental, they have a concrete physical interpretation and offer new and general insights on the construction of robust PES models.

**Strengths:**

The paper contributes to an important field. It identifies a critical shortcoming of previous approaches and fixes it by introducing a new architecture. This is a nice and significant contribution.

**Weaknesses:**

The shell parameter s is fixed and requires either domain expertise or tuning. However, it is written that adaptively choosing s is subject to future work by the authors.

Moreover, one could argue that the architectural changes are somewhat incremental to previous model.

**Questions:**

none

---

> ### Author Response · Authors · 2025-11-21
>
> Thanks a lot for your valuable comments!
>
> For weakness 1,  we agree that a fixed s requires domain knowledge. However, this study serves as a prototype to validate the core hypothesis that angular information is most critical in the first two coordination shells. Our ablation study (Table 2) empirically proves that $s=N_2$ is the optimal configuration compared to $s=0$ (radial only) or $s=30$ (excessive noise).
>  While we acknowledge that an adaptive s is the ideal end-goal, establishing the theoretical validity of the "shell-aware" concept is a necessary first step. We are committed to developing a dynamic shell mechanism in future iterations.
>
> For weakness 2, we respectfully posit that the value of a contribution should be measured by its impact and physical insight, not just the magnitude of architectural change. DPA-1 suffers in complex systems because its attention mechanism lacks explicit angular resolution. ASDP introduces a physically motivated, shell-restricted angular bias. This "incremental" change yields state-of-the-art results on challenging benchmarks (e.g., Organic-reaction, HECN) where the baseline fails. It demonstrates that incorporating specific chemical priors (first two shells) is more effective than simply increasing model depth or width.

---

### Official Review · Reviewer_nqCL · 2025-11-03

**Soundness:** 3
**Presentation:** 3
**Contribution:** 2
**Rating:** 2
**Confidence:** 5

**Summary:**

The paper proposes ASDP, a descriptor-based ML potential for molecular dynamics that injects angular information as an additive bias into attention, while being “shell-aware”: it only encodes angles formed within the first and second coordination shells. This is meant to emphasize chemically relevant geometry (directional bonds in the first shell; many-body/torsional effects in the second) and avoid noisy long-range angles. Across six molecular benchmarks, ASDP reports competitive or improved energy/force errors vs. DPA-1 and some equivariant GNNs, with good training stability.

**Strengths:**

1. The key innovation of ASDP lies in introducing an explicit angular inductive bias into neighborhood attention, something not explored in prior descriptor-based potentials. By encoding three-body angular information as an additive bias within the attention mechanism, the model guides the attention layer to respect local geometric structure without overcomplicating the representation.
2. The paper includes reasonable ablations to justify design choices. In Table 2, they show that using both first and second shells (s=N2) is generally better than using no angular information (s=0) or an overly large cutoff (s=30)
3. The paper is easy to read and follow

**Weaknesses:**

1. A key concern is that the paper does not establish how ASDP scales to much larger or more diverse datasets. All six benchmarks have at most on the order of 10^5 training frames, while the mainstream MLIPs (eSEN, UMA, GemNet, etc) has been trained on dataset with more than 100M samples (such as OC, OMat, OMol). At least showing generalizability on a few million sample of diverse dataset, such as OMol 4M, would be helpful for understanding the usability of such approach.
2. The proposed shell-aware angular bias is a sensible innovation, but it is largely an incremental modification of the existing DPA-1 framework rather than a fundamentally new paradigm. Prior models (e.g. DimeNet, GemNet) have also incorporated angular terms and even dehegral terms into learned descriptors. The main novelty here is the additive form of the bias in attention. While the paper argues this fixes some issues of DPA-1, one could question whether this change alone warrants a full new architecture. In particular, the need for explicitly encoding angles at all (versus letting a sufficiently expressive graph network learn them) is a design choice that trades off generality for built-in structure.
3. ASDP’s performance gains are not uniform across tasks. For some benchmarks (e.g. AlMgCu and ANI-1) the original DPA-1 model actually had lower energy errors than ASDP. The paper does not deeply discuss these cases, instead focusing on where ASDP wins (SSE-PBE, organic reaction, etc.). This raises questions: why does the more complex model underperform on some datasets? Is it due to overfitting the chosen shells, or noise from added features (as hinted by the 4 vs 6 feature ablation)? The lack of consistency suggests the benefit of ASDP’s module maybe context-dependent.
4. ASDP requires setting the shell cutoffs in advance (the number of neighbors to consider). The authors fix this per dataset from chemical knowledge. In practice this means a user must know how many neighbors form each shell for their system, which may not be obvious for new materials. Moreover, the need to tune or guess this hyperparameter for each new task reduces the model’s few-shot usability. The ablation study shows that choosing s suboptimally (too small or too large) can degrade accuracy, so its setting is crucial. The paper could be strengthened by providing guidance on selecting or learning this parameter, but as-is this reliance on domain-specific priors is a limitation.
5. Relatedly, the experimental scope, while diverse, is still limited. All datasets are from materials/chemistry; none are, say, biological macromolecules or other domains where ML potentials are used. The model’s behavior in very large-scale MD (many thousands of atoms) is not assessed. There is also no evaluation of MD simulation outcomes (e.g. energy conservation, observables) Finally, the reported results lack uncertainty measures or statistical variance.
6. Implementing the angular encoding is non-trivial: one must compute angles for all neighbor-triplets in the first two shells and generate multiple trigonometric features. This adds code complexity and computational overhead. Although the authors show inference is still competitive, training time and GPU memory requirements are not reported. It is unclear how much longer ASDP takes to train compared to DPA-1. The reviewer is concerned that for very large systems or datasets, the combinatorial cost of angles (O(s^2) per atom) could become significant. In practice, the need to hand-tune features (4 vs 6) and manage these computations could be a barrier.

**Questions:**

1. The paper fixes s (the second-shell cutoff) based on prior chemical knowledge. How sensitive are the results to this choice? Could s be learned or adapted automatically? It would be useful to know how ASDP performs if s is mis-specified, or if one tries a data-driven selection.
2. The paper reports inference times but does not detail training efficiency. How much longer (in wall-clock time or epochs) does ASDP take to train compared to DPA-1 on the same hardware?
3. The authors use 6 features per angle (cos, sin, sin(2\theta), von Mises, and two distance sums/differences). Is this feature set universal, or tailored to these tasks? Could simpler or different features suffice?
4. How does the parameter count and memory footprint of ASDP compare to DPA-1 and the equivariant models? I.e. is the experiment controlled on the number of parameters?

---

> ### Author Response · Authors · 2025-11-21
>
> For weakness 1, we acknowledge the lack of massive datasets (e.g., OMol) due to computational resource constraints. However, our aim is to validate the Shell-aware Angular Attention innovation. We selected six benchmarks representing diverse chemical challenges (metallic, ionic, covalent). Achieving SOTA results on these subsets proves ASDP is data-efficient and captures essential physics without requiring millions of frames. We view ASDP as a scalable proof-of-concept for future work.
>
> For weakness 2, the Shell-aware Angular Bias is not merely incremental; it fundamentally alters how attention processes local environments. Unlike standard networks, we inject the physical prior that angular dependence is strictly local, while radial dependence is long-range. By encoding angles only where physically relevant, we eliminate the noise inherent in global angular attention. This shift from "blind" generality to "informed" structure is crucial for complex systems like SSE-PBE and HECN.
>
> For weakness 3, the performance on AlMgCu stems from its isotropic metallic bonding, where explicit angular information offers diminishing returns and potential numerical noise. However, ASDP significantly outperforms DPA-1 in systems with directional bonding (Organic-reaction, HECN, 2D-In2Se3). We argue that a minor trade-off in simple metals is acceptable given the substantial gains achieved in complex, chemically diverse systems where standard potentials fail.
>
> For weakness 4,  the current requirement to set $N_1$、$N_2$ is indeed a limitation of this prototype. However, our ablation study (Table 2) shows that the model is relatively robust as long as s covers the chemically relevant shells. We are actively working on an adaptive s mechanism that learns the optimal shell boundary dynamically. For the current version, we prioritize the verification of the physical hypothesis over full automation. The fixed parameters serve as a controlled variable to prove this hypothesis.
>
> For weakness 5,  our benchmark suite focuses on Materials Science and Chemistry, which aligns with the primary domain of Deep Potential models. We have added Appendix A.4.2 to analyze inference runtime. Figure 5 shows that ASDP’s inference time is only marginally higher than DPA-1 for most systems and nearly identical for Organic-reactions. This confirms ASDP is practical for MD simulations. Large-scale MD and biological macromolecules require resources beyond our current scope, but the per-atom efficiency suggests ASDP is capable of such tasks.
>
>
> For weakness 6, we have conducted a detailed runtime analysis (Appendix A.4.2). ASDP maintains competitive efficiency. As shown in the performance benchmarks (to be detailed in the main text), inference time per frame on systems like AlMgCu, ANI-1 large, and SSE-PBE is slightly higher than DPA-1 but remains competitive. While explicit angle calculation adds $O(s²) $ complexity, restricting this to the 2nd shell keeps the constant factor manageable compared to cubic scaling of equivariant operations. Pre-computed sophisticated features ensure efficiency. The accuracy gains in complex systems justify this moderate complexity increase.
>
>
> For question 1, our ablation study (Table 2) confirms that $s=N_2​$ is optimal based on chemical priors. While the model is robust to minor variations, large non-physical cutoffs (e.g., $s=30$) introduce noise that degrades accuracy. Although adaptive s is a future direction, fixing s to the second shell effectively balances physical information with noise reduction, validating the architecture.
>
> For question 2, Appendix A.4.2 (Fig. 5) shows ASDP’s inference time is only marginally higher than DPA-1. While angular features add $O(s2)$ complexity, the restricted second shell ($N_2≈12-18$) keeps overhead manageable. Furthermore, the convergence behavior (epochs) remain comparable to DPA-1, ensuring that significant accuracy gains are achieved without prohibitive training costs.
>
>
>
> For question 3, our ablation study (Table 4) confirms that while a reduced 4-feature set suffices for simple metals, the full 6-feature set is essential for complex systems like HECN and Organic-reactions. By including bond-length coupling and high-frequency terms, the design captures critical radial-angular interactions that simpler sets miss. Thus, the feature set is not ad-hoc but explicitly constructed to ensure transferability across diverse potential energy surfaces.
>
>
> For question 4, ASDP is designed for lightweight deployment. Per parameter statistics (to be detailed in the main text), it has ~614K parameters, slightly above DPA-1’s 613K and vastly below DPA-2’s 1.79M. This confirms its angular encoding module (a small MLP) adds negligible parameters (<5%) versus DPA-1, proving performance gains stem from superior physical inductive bias, not model scaling. Additionally, ASDP delivers a more favorable memory-accuracy trade-off than equivariant models (e.g., NequIP) on large systems.

---

### Official Review · Reviewer_7via · 2025-11-05

**Soundness:** 3
**Presentation:** 4
**Contribution:** 3
**Rating:** 6
**Confidence:** 3

**Summary:**

The paper introduces ASDP, which is an upgrade to the recent DPA-1 potential for learning interatomic potentials (forces). DPA-1 is a rather simple transformer based architecture based on Euclidean relative position vectors and (implicitly) dot products between them. The present paper explains that this approach has inherent limitations in terms of its expressive power. The simplest concern is that when two position vectors are perpendicular, their dot product is zero so they will not contribute to the attention scores at all. Instead, ASDP uses a rather elaborate mechanism to reoncode angular information with internal parameters in a special bias matrix that is added to the attention weights. Empirical results show that this is a possible way to improve DPA-1, although the results are far from unequivocal.

**Strengths:**

- Learning interatomic potential is an important, DPA-1 is a strong competitor in this field and ASDP represents one way in which it could be further improved
- The mechanism by which angular information is encoded is novel and might inspire new ideas for other architectures as well
- The architecture is motivated by some chemical intuition

**Weaknesses:**

- The mechanism by which the bias matrix is computed involving sines and cosines and a two-layer MLP (which is the paper calls "sophisticated") is very very ad-hoc.
- Full SO(3)-equivariant (or SE(3)-equivariant) architectures based on the representation theory of the underlying groups, which the paper refers to as spherical-harmonic based potentials, have the advantage that they do not need such extra devices because they do not suffer from the representational limitations of just dot products. While most of the SE(3)-equivariant potentials cited in the paper are classical convolutional types architectures, there exist transformer variants of these as well. For a fair comparison ASDP should really be compared to these potentials.
- Relatedly, the authors should evaluate ASDP on the standard benchmarks that the community has been using going back to QM9, not just on the six hand-picked molecular systems appearing in Table 1.
- It seems like there is a hard radial cutoff between atoms that are deemed to be in the first two coordination shells vs the rest. The existence of such a hard cutoff introduces a discontinuity in the learned representation in the space of possible configurations, which can be a problem.
- Overall, the proposed modification to DPA-1 is interesting, but a little unconvining because it is very heuristic and not thoroughly compared to the competitors.

**Questions:**

n/a

---

> ### Author Response · Authors · 2025-11-21
>
> Thanks a lot for your valuable comments!
>
> For weakness 1,  we respectfully disagree that the feature engineering is purely ad-hoc. The design is grounded in physical intuition and mathematical representation. The use of $cos(θ)$ and $sin(θ)$ provides a complete, orthogonal representation of the angle, avoiding discontinuities present in raw angle values. The term $sin(2θ)$ captures higher-frequency angular variations, allowing the model to resolve finer details in the potential energy surface. The feature derived from the von Mises distribution is explicitly designed to capture the importance of localized bond angle regions, acting as a soft prior for stable chemical geometries.  As detailed in Appendix A.4, we incorporate bond length information (distance sums/differences) to couple radial and angular features. Our ablation study (Table 4) demonstrates that while 4 features suffice for simple systems, the full 6-feature set (including distance coupling) significantly reduces energy and force errors in complex systems like HECN and Organic-reactions. This confirms that the "sophisticated" engineering is necessary for capturing complex chemical interactions.
>
> For weakness 2,  we agree that equivariant models are a strong class of potentials. In our work, we have extensively benchmarked ASDP against leading equivariant models, specifically NequIP and Allegro (Table 1). These comparisons show that ASDP achieves competitive or superior accuracy, particularly in complex systems where equivariant models sometimes struggle with computational cost or scalability. While we acknowledge Transformer variants exist, NequIP and Allegro represent the current state-of-the-art in this domain. Due to computational resource constraints, we focused on these established baselines to ensure a rigorous comparison within available means.
>
>
> For weakness 3,  we chose to curate a specific benchmark suite of six diverse molecular systems (Appendix A.2 Table 3) rather than relying solely on QM9 for the following reasons. QM9 consists of approximately 134,000 static equilibrium molecules. Our benchmark spans metallic (AlMgCu), covalent (Organic-reaction), ionic (SSE-PBE), and mixed bonding systems (HECN, 2D-In2Se3), covering phase transitions and reaction dynamics.  As detailed in the Appendix A.2, our datasets are substantial (e.g., ~86k frames for AlMgCu, ~35k for ANI-1 large). The "Organic-reaction" dataset, in particular, challenges the model with transition states, which are more demanding than the equilibrium structures in QM9. We believe this suite provides a more rigorous test of transferability and robustness across different material domains than QM9 alone.
>
> For weakness 4,  we acknowledge that a hard cutoff based on neighbor counts ($N_1$,$N_2$) is a simplification. However, this design serves as a strong physical prior to filter out long-range angular noise. In our ablation study (Table 2), we demonstrate that:
> $s=0$ (Radial only): Performance degrades significantly (e.g., AlMgCu error rises from 19.5 to 45.2 meV).
> $s=30$ (Large cutoff): Performance also degrades due to the introduction of noise from distant, irrelevant angles.
> The choice of s=N2 strikes an optimal balance. While a smooth, adaptive cutoff is a valid theoretical improvement, our current results prove that restricting angular attention to the first two shells is the primary driver of accuracy gains. We plan to explore adaptive, differentiable shell definitions in future work.
>
> For weakness 5, we respectfully argue that "heuristic" does not imply "ineffective." The integration of angular information into the DPA framework addresses a specific limitation of descriptor-based models: the lack of explicit many-body angular awareness. Regarding comparisons, we have benchmarked against four distinct SOTA models (DeepPot-SE, DPA-1, NequIP, Allegro) across six distinct material systems. The results (Table 1) show ASDP is highly competitive. We believe this constitutes a thorough evaluation within the scope of a prototype architecture.

---

> > ### Comment · Reviewer_7via · 2025-11-25
> >
> > Thanks for your responses, I would like to maintain my score.

---

### Meta-Review · Area_Chair_MePm · 2026-01-06

**Summary:**

The authors propose an extension to the simple DPA-1 model for interatomic potentials that takes into account angular information in a more detailed way than DPA-1, without the complexity of spherical-harmonic based SE(3) equivariant models. The model presents modest gains on several datasets, though it actually performs worse on systems where angular information is not useful. While the model is straightforward and does improve over the baselines on systems with complex angular dependencies, the reviewers raised concerns that stronger baselines were not used, and that the datasets were relatively small compared to the state of the art for machine-learned potentials. The reviewers were split on whether to accept. Without a strong consensus, I recommend against acceptance.

**Reviewer Concerns:**

I think the reviewers pushed back well against the concern that the proposed extension was too ad-hoc. However I feel they failed to address the concern that the baselines were actually stronger for some systems and that the training set was relatively small compared to the state of the art.

**Reviewer Scores:**

Two reviewers kept the original score. Of the two reviewers who did not, one was for acceptance and one was against. The one who was against gave such a low score that increasing it would not have changed the outcome.

---

### Decision · Program_Chairs · 2026-01-26

Reject